# Polyphasic Characterization and Genomic Insights into an Aerobic Denitrifying Bacterium, *Shewanella zhuhaiensis* sp. nov., Isolated from a Tidal Flat Sediment

**DOI:** 10.3390/microorganisms11122870

**Published:** 2023-11-27

**Authors:** Yang Liu, Tao Pei, Juan Du, Honghui Zhu

**Affiliations:** Key Laboratory of Agricultural Microbiomics and Precision Application (MARA), Guangdong Provincial Key Laboratory of Microbial Culture Collection and Application, Key Laboratory of Agricultural Microbiome (MARA), State Key Laboratory of Applied Microbiology Southern China, Institute of Microbiology, Guangdong Academy of Sciences, Guangzhou 510070, China; liuhaolin3@163.com (Y.L.); a698921pei@163.com (T.P.); dujuan@gdim.cn (J.D.)

**Keywords:** *Shewanella zhuhaiensis*, taxonomy, novel species, genome functional analysis, phylogenetic analyses

## Abstract

A new, facultatively anaerobic, light-yellow, and rod-shaped bacterium designated as 3B26^T^ isolated from Qi’ao Island’s tidal flat sediment was identified. Strain 3B26^T^ can hydrolyze gelatin, aesculin, and skim milk. The major cellular fatty acids were identified as iso-C_15:0_, referred to as summed feature 3, and C_16:0_; the polar lipids consisted of phosphatidylethanolamine, phosphatidylglycerol, and phospholipid; and the quinones contained Q-7, Q-8, MK-7, and MMK7. The genomic size of strain 3B26^T^ was 4,682,650 bp, and its genomic DNA G + C content was 54.8%. While a 16S rRNA gene-based phylogenetic analysis confirmed that strain 3B26^T^ belongs to the genus *Shewanella*, both phylogenomic inference and genomic comparison revealed that strain 3B26^T^ is distinguishable from its relatives, and digital DNA-DNA hybridization (dDDH) values of 24.4–62.6% and average nucleotide identities (ANIs) of 83.5–95.6% between them were below the 70% dDDH and 96% ANI thresholds for bacterial species delineation. Genomic functional analysis demonstrated that strain 3B26^T^ possesses complete gene clusters of eicosapentaenoic acid biosynthesis and denitrification. Based on the evidence above, strain 3B26^T^ is considered to represent a novel species of the genus *Shewanella*, and the name *Shewanella zhuhaiensis* sp. nov. (type strain 3B26^T^ = GDMCC 1.2057^T^ = KCTC 82339^T^) is proposed.

## 1. Introduction

*Shewanella* is the largest genus of the family *Shewanellaceae* [1] of the order *Alteromonadales* within the class *Gammaproteobacteria* and was first established by MacDonell et al. [2]. Since the second half of the 1990s, the published species number in the genus *Shewanella* has grown steadily [3]. Until now, the genus *Shewanella* has comprised nearly 100 species with validly published names listed in the LPSN [4], highlighting its taxonomic diversity. The genus *Shewanella* is characterized by its metabolic versatility, impressive global distribution, and great application potential, such as probiotics, bioremediation, and anti-tumor metabolites [5]. More significantly, strains of some species within the genus *Shewanella* have become emerging infectious pathogens worldwide and are harmful to humans, aquatic livestock, and food, mainly including *Shewanella algae*, *Shewanella putrefaciens*, and *Shewanella xiamenensis* [6]. Therefore, it is of great importance to isolate and identify novel *Shewanella* strains from diverse environments.

In the process of mining cultivatable bacterial strains from nearshore sediments, a light-yellow isolate designated as 3B26^T^ was obtained. This report aims to identify the taxonomic status of the novel isolate designated as 3B26^T^, isolated from a tidal flat sediment. The polyphasic taxonomic analyses indicated that this novel isolate should be considered a novel species of the genus *Shewanella*, designated as *Shewanella zhuhaiensis* sp. nov.

## 2. Materials and Methods

### 2.1. Bacterial Isolation

Strain 3B26^T^ was isolated from Qi’ao Island’s tidal flat sediment (N 22°24′41″, E 113°38′50″), Zhuhai, Guangdong, P. R. China (Appendix A). The sample treatment, bacterial isolation, and preservation were performed according to the methods in [7], with a difference in marine agar (MA, BD) as the isolation medium. Among all isolates, a strain with a light-yellow color designated as 3B26^T^ was used for the taxonomic analysis in this study. Strain 3B26^T^ was maintained in 25% (*v*/*v*) glycerol suspension at −80 °C for long-term preservation. The most closely related type strains, *Shewanella amazonensis* SB2B^T^ (=CIP 105786^T^) [8], *Shewanella litorisediminis* SMK1-12^T^ (=KCTC 23961^T^) [9], and *Shewanella jiangmenensis* JM162201^T^ (=GDMCC 1.2006^T^) [10], were obtained from the Institut Pasteur Collection (Paris, France), the Korean Collection for Type Cultures (Jeongeup-si, Korean), and the Guangdong Microbial Culture Collection Center (Guangzhou, China), respectively. Given national regulations on the import and export of pathogens, the type strain *Shewanella khirikhana* TH2012^T^, as a shrimp pathogen, currently cannot be obtained from culture collections or authors, and thus, its taxonomic data were acquired based on species identification [11]. Since the species *Shewanella zhangzhouensis* was recently published, the data on its type strain, FJAT-52072^T^, was from the study described in [12]. The physiological and biochemical identification and enzyme activity tests (casein and gelatin hydrolysis, oxidase, and lipase) of strain 3B26^T^ and the reference type strains were carried out on MA or in marine broth (MB, BD) under identical culture conditions.

### 2.2. Morphological, Physiological, and Biochemical Characteristics

The colony and cellular morphological characteristics and flagella formation of strain 3B26^T^ were observed using a stereomicroscope (SZX10, Olympus^®^, Tokyo, Japan) and a transmission electron microscope (TEM, H7650, Hitachi, Tokyo, Japan), respectively. A Gram reaction was performed using a Gram staining kit (Hope, Qingdao, China), following instructions. The enzymatic activities of catalase and oxidase; cellular motility; and growth tests on different media, temperatures, pHs, and NaCl concentrations were determined following methods in [7]. A test of the anaerobic growth of strain 3B26^T^ was conducted with an anaerobic pouch (MGC, Mitsubishi, Tokyo, Japan) on MA at an optimal growth temperature for 4 weeks. The hydrolysis of starch; casein; carboxymethyl cellulose; skim milk; and Tweens 20, 40, 60, and 80 was tested according to the method in [7]. Additional enzymatic activities and physiological and biochemical properties were examined using API ZYM and API 20NE (Bio-Mérieux, Craponne, France) strips based on the manufacturer’s instructions.

Cells of strain 3B26^T^ and the reference type strains were grown on the third quadrants of the MA up to the exponential growth phase and were collected and then freeze-dried by using a vacuum freeze-drying apparatus. The saponification, methylation, and extraction of harvested cells were conducted according to the protocol of the Sherlock^®^ Microbial Identification System (MIS; MIDI, Inc., Newark, DE, USA). The extraction and separation of the polar lipids of strain 3B26^T^ were conducted using a chloroform–methanol system [13] and two-dimensional thin-layer chromatography with silica gel 60 F254 (Merck, Darmstadt, Germany), respectively. Analyses of the fatty acids and polar lipids obtained were conducted following the methods in [7]. Quinones were identified using HPLC (Agilent 1200, Santa Clara, CA, USA; ODS 250 × 4.6 mm × 5 µm; flowing phase methanol–isopropanol of 2:1 at 1 mL·min^−1^) according to the Collins method [14].

### 2.3. The 16S rRNA Gene Sequencing and Phylogenetic Analysis

The genomic DNA of strain 3B26^T^ was isolated by using a commercial kit (Magen, Guangzhou, China), following instructions. The 16S rRNA gene amplification and sequencing of strain 3B26^T^ were conducted using the primer pair 27F/1492R [15] via PCR. The 16S rRNA gene comparison of strain 3B26^T^ was carried out using the recently updated EzBioCloud database [16]. The identities of the 16S rRNA gene sequence of strain 3B26^T^ and the closely related type strains were compared using the software DNAMAN version 8. Multiple alignments of 16S rRNA gene sequences were performed using the Muscle algorithm implemented in the software MEGA version 11.0.13 [17]. The 16S rRNA gene-based phylogenetic trees were reconstructed with MEGA using the three usual methods with default parameters: neighbor-joining (NJ) [18], maximum likelihood (ML) [19], and maximum evolution (ME) [20]. The Kimura 2-parameter and Tamura–Nei methods were, respectively, chosen as substitution models for the NJ and ML methods. The robustness of the three phylogenetic trees was assessed using a 1000-replicate bootstrap analysis [21]. Type strain *Alteromonas macleodii* ATCC 27126^T^ was used as an outgroup in the 16S rRNA gene-based phylogenetic analysis.

### 2.4. Whole-Genome Sequencing, Phylogenomic, and Comparative Analyses

The whole genome of strain 3B26^T^ was sequenced using the Illumina NovaSeq PE150 platform (Majorbio, Guangzhou, China). High-quality reads were assembled with the software SPAdes version 3.8.1 with default parameters [22]. The available genomes of all type strains within the genus *Shewanella* were downloaded from the GenBank database. Given that most genomes were in draft status, their quality was assessed with the software CheckM version 1.1.2 [23]. The digital DNA-DNA hybridization (dDDH) values between genome sequences of strain 3B26^T^ and closely related type strains were calculated using Genome-to-Genome Distance Calculator version 2.1, an online service, with the recommended formula 2 and BLAST+ local alignment tool [24]. The average nucleotide identity (ANI) values of the genome sequences were scored using the software FastANI version 1.31 [25]. To maintain the consistency of the genomic comparative analysis, the gene prediction and annotation of strain 3B26^T^ and its close relatives were re-conducted with the software Prokka version 1.13 [26] and the Rapid Annotation using Subsystem Technology (RAST) server [27] with default parameters. The functional classification of the predicted genes was carried out with clusters of orthologous Ggoups (COG) annotations using eggNOG mapper v2 [28]. The 92-bacteria core gene-based phylogenomic tree was reconstructed with the software UBCG version 3.0 (www.ezbiocloud.net/tools/ubcg, accessed on 24 October 2023) [29] following the default pipeline. Type strain *Alteromonas macleodii* ATCC 27126^T^ was also used as an outgroup in the phylogenomic analysis.

### 2.5. Accession Numbers

The 16S rRNA gene and genome sequences of strain 3B26^T^ were submitted to the NCBI database with the accession numbers OM761198 and JAKUDL000000000.

## 3. Results and Discussion

### 3.1. Physiological Characterization

Visible colonies of strain 3B26^T^ were light-yellow-colored on MA for 48 h when incubated at 30 °C, and its cells were observed to be straight or curved in a rod shape and motile by means of a single polar flagellum (Figure 1). As shown in Table 1, the colony colors of strain 3B26^T^ and its relatives were significantly different and thus could be used as a simple indicator to distinguish them [8,9,10,11,12]. The cell growth of strain 3B26^T^ was found to occur at 10–40 °C (optimum, 28 °C) and at pH 4.0–9.0 (optimum, 7). The cells could tolerate up to 5% NaCl (*w*/*v*), as growth was not observed at 6% NaCl. The cells could grow on MA, R2A, LB, TSA, and NA media. Strain 3B26^T^ was positive for oxidase and catalase and could hydrolyze skim milk. Strain 3B26^T^ and the reference type strains could reduce nitrate to nitrite, except for *S*. *zhangzhouensis* FJAT-52072^T^ [12]. The cells were found to be Gram-stain-negative (Figure 1) and positive for catalase; oxidase; the hydrolysis of Tweens 20, 40, 60, and 80; skim milk; gelatin; and aesculin but negative for the hydrolysis of starch, casein, and carboxymethyl cellulose. Unlike strain 3B26^T^, type strain *S*. *Khirikhana* TH2012^T^ could hydrolyze casein [11]. The cells were found to assimilate D-glucose, D-maltose, N-acetyl-glucosamine, D-mannose, and malic acid as carbon sources. API 20NE tests indicated that the profiles of the substrate utilization of strain 3B26^T^ and its relatives were different, especially in D-glucose, D-mannose, and potassium gluconate (Table 1). The physiological characteristics of strain 3B26^T^ were similar to those of related reference type strains (Table 1). Nevertheless, some distinct characteristics of strain 3B26^T^ in Table 1 can be distinguished from its related type strains.

### 3.2. Chemotaxonomic Analysis

The cellular fatty acid compositions of strain 3B26^T^ and its closely related type strains, including *S. zhangzhouensis* FJAT-52072^T^, *S. amazonensis* SB2B^T^, *S. litorisediminis* SMK1-12^T^, and *S. jiangmenensis* JM162201^T^, are provided in Table 2. The major fatty acids (more than 10% of total fatty acids) of strain 3B26^T^ were iso-C_15:0_ (16.5%), summed feature 3 (C_16:1_ *ω*6*c* and/or C_16:1_ *ω*7*c*, 14.1%), and C_16:0_ (11.0%). The minor fatty acids (5–10% of total fatty acids) were C_17:1_ *ω*8*c* (8.5%) and iso-C_13:0_ 3OH (5.3%). The prominent fatty acids of strain 3B26^T^ were similar to those of the reference type strains, indicating that strain B26^T^ should be considered a member of the genus *Shewanella*. However, strain B26^T^ contained a higher amount of iso-C_13:0_ 3OH and iso-C_16:0_ than its closely related type strains, which could make it distinguishable from close relatives. The polar lipids of strain 3B26^T^ contained phosphatidylethanolamine (PE), phosphatidylglycerol (PG), and phospholipid (PL) (Appendix A). The polar lipid profile of strain 3B26^T^ was similar to those of *S. zhangzhouensis* FJAT-52072^T^ [12] and *S. jiangmenensis* JM162201^T^ [10] in that PE and PG were the main components. Strain 3B26^T^ simultaneously included both ubiquinones (Q-7 and Q-8, 63.6% and 36.4%) and menaquinones (MK-7 and MMK7, 79.9% and 20.1%), which was in line with related reference type strains within the genus *Shewanella* [9,10].

### 3.3. Genome Structural Features

The completeness and contamination of the final assembled genome of strain 3B26^T^ were 100% and 0.62% based on the estimation performed by CheckM (Appendix A). The complete 16S rRNA gene sequence of strain 3B26^T^ was extracted from its assembled genome and shares 100% of its identity with that obtained via PCR, confirming the authenticity of the genome assembly. Therefore, the genome of strain 3B26^T^ can be considered high quality. The genome of strain 3B26^T^ consists of 18 contigs with a genome size of 4,682,650 bp and genome DNA G + C content of 54.8% (Figure 2). Similar to strain 3B26^T^, the genomes of the reference type strains were of high quality given the ≤1% contamination and 100% completeness [23] determined by the CheckM. The genomic size of strain 3B26^T^ was smaller than that of *S*. *khirikhana* TH2012^T^ and greater than those of the other type strains. The genomic DNA G + C content of strain 3B26^T^ (54.8%) was higher than those of *S*. *amazonensis* SB2B^T^ (53.6%), *S. zhangzhouensis* FJAT-52072^T^ (53.7%), and *S*. *litorisediminis* SMK1-12^T^ (54.0%) and slightly lower than those of *S. khirikhana* TH2012^T^ (54.9%) and *S. jiangmenensis* JM162201^T^ (55.0%). In the genome sequence of strain 3B26^T^, 4083 CDSs, 3 rRNA genes (a gene copy of 5S rRNA-16S rRNA-23S rRNA), and 64 tRNA genes were annotated (Appendix A).

### 3.4. Phylogenetic and Taxonogenomic Analyses

The 16S rRNA gene sequence of strain 3B26^T^ with 1412 bp obtained via PCR was compared with those of all species within the genus *Shewanella*. The EzBioCloud server (www.ezbiocloud.net, accessed on 20 October 2023) was used to search for its closest type strains. The results show that strain 3B26^T^ is closely related to members of the genus *Shewanella*, sharing the closest identity with *S*. *khirikhana* TH2012^T^ (99.5%), followed by *S. zhangzhouensis* FJAT-52072^T^ (99.0%), *S*. *litorisediminis* SMK1-12^T^ (99.0%), and *S. amazonensis* SB2B^T^ (98.6%) and below 97.0% with other members of this genus. The inference of the 16S rRNA gene-based phylogenetic trees showed that strain 3B26^T^ fell into the genus *Shewanella* and has the closest phylogenetic relationship with *S*. *khirikhana* TH2012^T^, as shown in the NJ trees (Figure 3). As shown in Appendix A, the 16S rRNA gene-based ML and ME phylogenetic trees presented similar phylogenetic relationships between strain 3B26^T^ and its close relatives above. As a result, the 16S rRNA gene-based analysis indicated that strain 3B26^T^ is a member of the genus *Shewanella*.

Phylogenomic inference has proved to be a reliable method of determining species delineation in bacterial taxonomy [30]. The dDDH values between strain 3B26^T^ and reference type strains *S*. *khirikhana* TH2012^T^, *S. zhangzhouensis* FJAT-52072^T^, *S*. *amazonensis* SB2B^T^, *S*. *litorisediminis* SMK1-12^T^, and *S*. *jiangmenensis* JM162201^T^ were 62.6, 25.6, 25.2, 24.7, and 24.4%, respectively. These dDDH values were well below 70% dDDH, which is a recognized threshold for bacterial species delineation [31]. The ANI values between strain 3B26^T^ and its relatives above were 95.6, 84.4, 84.2, 83.9, and 83.5%, respectively. Since 2009, the 95–96% ANI has always been used as a prokaryotic species boundary [32]. However, increasing studies have suggested that the 70% dDDH value is not equivalent to the 95–96% ANI but instead approximately 96.0% or slightly higher [33,34,35]. In this study, the comparative analysis of dDDH and ANI values proved once again that the ANI threshold for species delineation should be raised to around 96%. Moreover, the genome-based phylogenetic analysis indicated that strain 3B26^T^ formed a separate branch distinguishable from related reference type strains with a 100% bootstrap value (Figure 4). Therefore, the genome-based analysis supported the proposal that strain 3B26^T^ represents a novel genospecies of the genus *Shewanella*.

### 3.5. Genomic Functional Analysis

As per RAST’s annotation and 8ggnog’s functional classification, 296 genes for energy production and conversion (abbreviated as category C, the same as below), 289 genes for transcription (K), 282 genes for cell wall/membrane/envelope biogenesis (M), 268 genes for signal transduction mechanisms (T), and 267 genes for amino acid transport and metabolism (E) were identified in strain 3B26^T^ (Figure 5). Overall, the genome-wise functional comparison showed that the gene number and distribution pattern of the COG categories of strain 3B26^T^, *S*. *khirikhana* TH2012^T^, *S. zhangzhouensis* FJAT-52072^T^, *S*. *amazonensis* SB2B^T^, *S*. *litorisediminis* SMK1-12^T^, and *S*. *jiangmenensis* JM162201^T^ were similar, and most predicted genes belonged to several of the central biological functional categories above, which are essential to survival (Appendix A). Polyunsaturated fatty acids, such as *ω*-3 PUFAs, including eicosapentaenoic acid (EPA) and the counterpart docosahexaenoic acid, play essential roles in signaling processes and membrane biology in most living organisms. In this study, the EPA synthesis gene clusters (*pfaABCDR*-*pfaE*) were identified in the genomes of strain 3B26^T^ and its reference type strains. The EPA could help these strains adapt to cold environments [36].

Central metabolic pathways including nitrogen, sulfur, and phosphorus metabolism were also compared between strain 3B26^T^ and its reference type strains based on the RAST annotation and the local BLAST (Table 3). For nitrogen metabolism, denitrification is a process that converts nitrate into nitrogen gas. The genes encoding for periplasmic nitrate reductase (Nap), nitrite reductase (Nir), and nitric oxide reductase (Nor) were found in the genomes of strain 3B26^T^ and five reference type strains, except for Nir in *S. jiangmenensis* JM162201^T^, which was absent. Intriguingly, all strains possessed two periplasmic nitrate reductase systems, Nap-*α* and Nap-*β*, which is a special trait of *Shewanella* bacteria [37,38]. The genome of strain 3B26^T^ contained a complete pathway for dissimilatory nitrate reduction (Nap, Nir, and Nrf), while those of five of the reference type strains lacked the genes encoding for nitrite reductase (NirBD). The genes encoding for assimilatory nitrate reductase (NasAB) were only found in strain 3B26^T^. The above-mentioned multiple nitrogen metabolic pathways are likely to provide competitive advantages for themselves in complicated environments for survival. For sulfur metabolism, all strains possessed a complete pathway for assimilatory sulfate reduction (CysNDCHJI), indicating that they can reduce sulfate into hydrogen sulfide [39]. For phosphorus metabolism, all strains included two two-component systems of PhoR-PhoB and PgtABC; the former plays a crucial role in sensing and responding to changes in environmental phosphate concentrations [40], and the latter controls the expression of the phosphoglycerate transport system [41]. The high-affinity inorganic phosphate transporter of PstSCAB, the low-affinity inorganic phosphate transporter of Pit, and the phosphonate transport of PhnCDE were found in the genomes of all strains, which enables these strains to utilize inorganic phosphate more efficiently from the environment, especially when phosphorus is limited. The genes encoding for alkaline phosphatases PhoA, PhoD, and PhoX were found in all strains and encoded for an acid phosphatase PhoN only found in *S*. *jiangmenensis* JM162201^T^, which suggested that organic phosphorus can also be utilized as a phosphorus source by these strains. As a result, the genomic functional analysis of strain 3B26^T^ and its reference type strains demonstrated their metabolic versatility.

The colony colors of strain 3B26^T^ and its reference type strains ranged from yellow to red, as shown in Table 1. A study by Coon et al. demonstrated that tyrosine, its structural analogs, or its metabolites may serve as precursors for melanin biosynthesis, and the melanin is usually yellow, brown, or red [42]. Inspired by this study, we reasonably speculated that these strains are highly likely to synthesize melanin. An analysis of the whole-genome sequences of these strains showed that they contained the *phhA*, *tyrB*, and *hppD* genes, which are responsible for the sequential conversion of phenylalanine and tyrosine into 4-hydroxyphenyl pyruvate and homogentisic acid (HGA). HGA accumulates and undergoes auto-oxidation and polymerization, resulting in melanin production [43]. In general, HGA can also be further degraded into fumarate and acetoacetate under the action of multiple enzymes encoded by *hmgABC* genes [44]. However, the genome sequences of strain 3B26^T^ and its reference type strains did not contain *hmgBC* genes (unpublished data). In this case, HGA accumulated and further converted into melanin-like yellow, orange, and red pigments.

## 4. Conclusions

A polyphasic taxonomic approach including phylogenetic inferences based on 16S rRNA genes and genomes; examinations of morphological, biochemical, and physiological characteristics; and chemotaxonomic properties provided sufficient and reliable evidence to differentiate strain 3B26^T^ from its closest-related type strains and to further confirm that it represents a novel species of the genus *Shewanella*. We propose the name *zhuhaiensis* sp. nov. for this novel member of the genus *Shewanella*, with 3B26T as the type strain. A description of *Shewanella zhuhaiensis* sp. nov. *Shewanella zhuhaiensis* (zhu.hai.en’sis. N.L. fem. adj. *zhuhaiensis*, from Zhuhai, China, where the strain was isolated) can be found below.

Its cells are Gram-stain-negative, facultatively anaerobic, straight or curved in a rod shape with 0.4–0.6 μm in width and 2.0–2.8 μm in length, and motile by means of a single polar flagellum. Its cells can grow on MA, R2A, LB, TSA, and NA media. Its colonies are round, smooth, and light-yellow-colored. Growth can be observed at 10–40 °C with an optimum of 30 °C at pH 4.0–9.0 with an optimum of 7.0 and in 0–5.0% NaCl with an optimum of 1.0% (*w*/*v*). It is positive for catalase; oxidase; the hydrolysis of skimmed milk; and Tweens 20, 40, 60, and 80, and it is negative for the hydrolysis of starch, casein, and carboxymethyl cellulose. In API ZYM tests, enzymatic activities, including alkaline and acid phosphatase, esterase lipase (C8), lipase (C14), esterase (C4), valine arylamidase, leucine arylamidase, trypsin, cystine arylamidase, *α*-chymotrypsin, N-acetyl-*β*-glucosaminidase, and napthol-AS-BI-phosphohydrolase were positive; the remaining were negative. In API 20NE tests, nitrate can be reduced to nitrite; aesculin and gelatin can be hydrolyzed; and D-glucose, D-maltose, D-mannose, N-acetyl-glucosamine, and malic acid can be utilized; the remaining are negative. iso-C_15:0_, referred to as summed feature 3 (C_16:1_ *ω*6*c* and/or C_16:1_ *ω*7*c*), and C_16:0_ were identified as its major cellular fatty acids. Phosphatidylethanolamine and phosphatidylglycerol are its predominant polar lipids. The ubiquinones of Q-7 and Q-8 and the methylquinones of MK-7 and MMK7 were detected. The G + C content of the genomic DNA is 54.8%.

The type strain designated as 3B26^T^ (=GDMCC 1.2057^T^ = KCTC 82339^T^) was isolated from Qi’ao Island’s tidal flat sediment in Zhuhai, Guangdong, P. R. China. The GenBank/EMBL/DDBJ accession numbers of the 16S rRNA gene and genome sequences of strain 3B26^T^ are OM761198 and JAKUDL000000000, respectively.

## Figures and Tables

**Figure 1 microorganisms-11-02870-f001:**
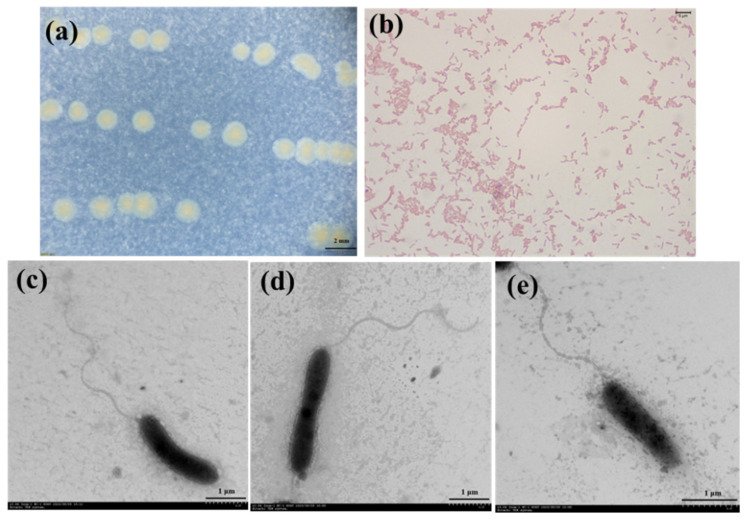
Colony morphology (**a**), Gram-stained cells (**b**), and cellular morphology (**c**–**e**) of strain 3B26^T^.

**Figure 2 microorganisms-11-02870-f002:**
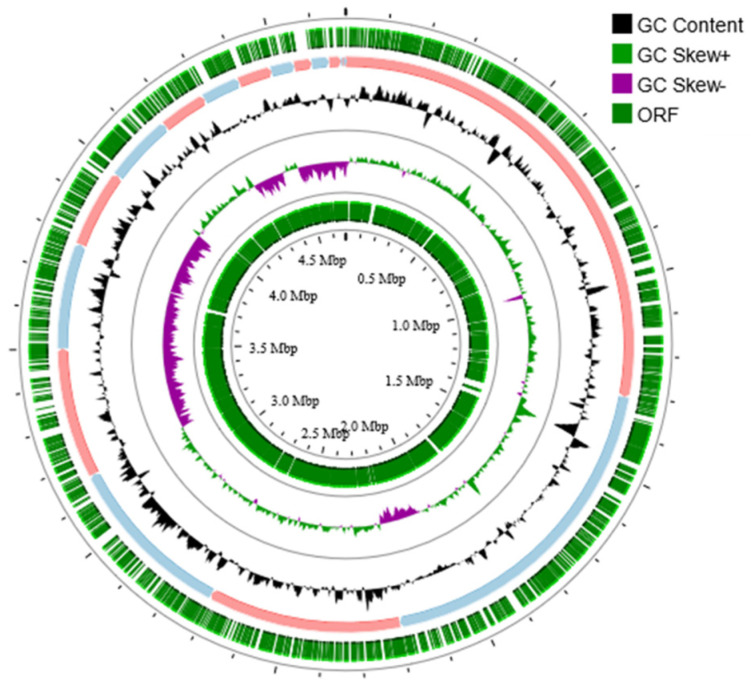
Circular genome map of strain 3B26^T^. Circles from the outside to inside show ORF, Contig 1-18, GC content, and skew.

**Figure 3 microorganisms-11-02870-f003:**
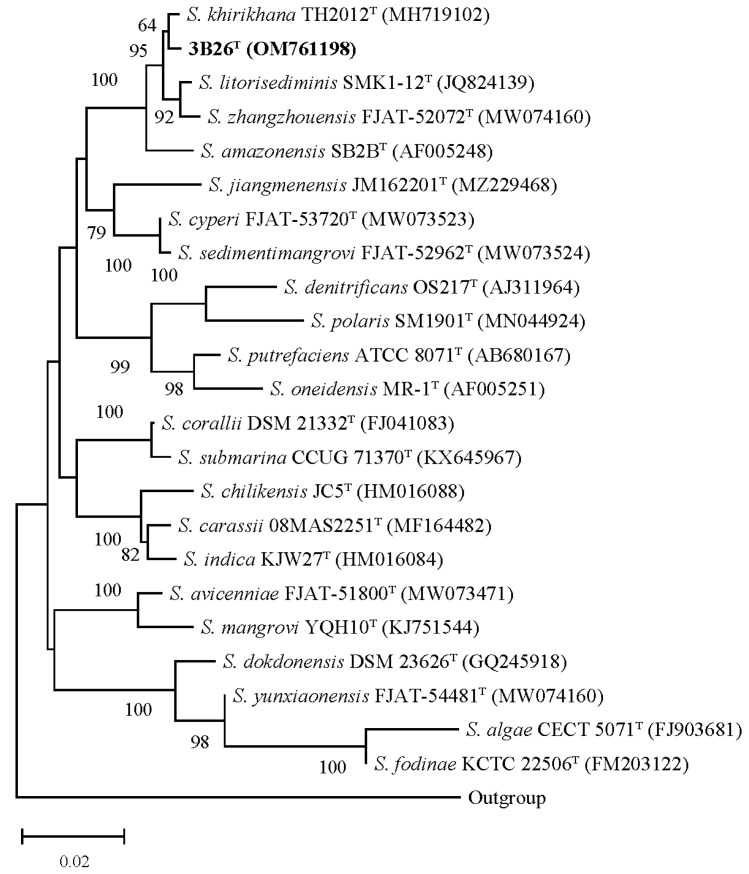
The neighbor-joining tree derived from 16S rRNA gene sequences with a common region including more than 1300 bp shows the phylogenetic relationships between strain 3B26^T^ and its related reference type strains. Type strain *Alteromonas macleodii* ATCC 27126^T^ (accession number of its 16S rRNA gene: Y18228) is used as an outgroup. More than 60% of the bootstrap values are shown at branch points. Bar: 0.02 represents the number of substitutions per nucleotide position. The accession numbers for the 16S rRNA genes are shown in parentheses.

**Figure 4 microorganisms-11-02870-f004:**
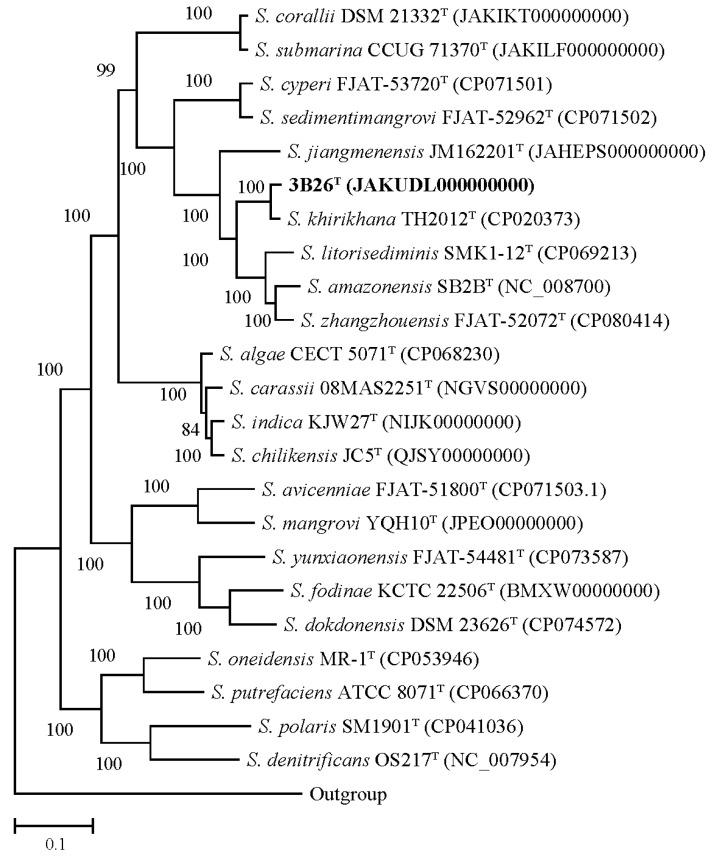
The core gene-based phylogenomic tree shows the phylogenetic relationships between strain 3B26^T^ and its related type strains. More than 80% of the bootstrap values are shown at branch points. Type strain *Alteromonas macleodii* ATCC 27126^T^ (accession number of its genome: NC_018632) is used as an outgroup. Bar: 0.1 represents the number of substitutions per site. The accession numbers of the genomes of the type strains are shown in parentheses.

**Figure 5 microorganisms-11-02870-f005:**
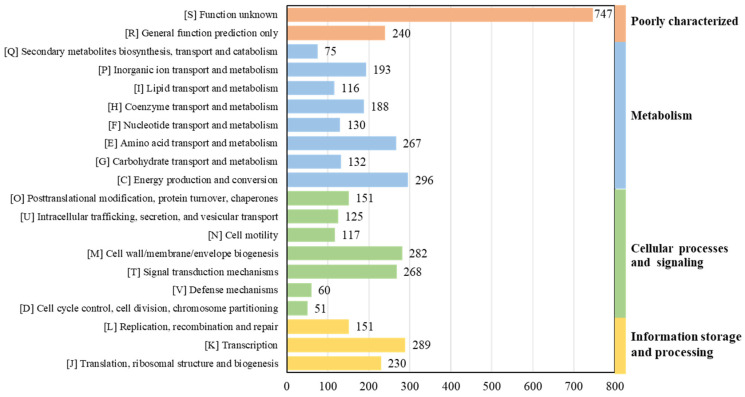
The gene distribution of COG categories in strain 3B26^T^.

**Table 1 microorganisms-11-02870-t001:** Differential characteristics between strain 3B26^T^ and closely related reference type strains within the genus *Shewanella*. Strains: 1, 3B26^T^; 2, *S*. *khirikhana* TH2012^T^; 3, *S. zhangzhouensis* FJAT-52072^T^; 4, *S. amazonensis* SB2B^T^; 5, *S. litorisediminis* SMK1-12^T^; 6, *S. jiangmenensis* JM162201^T^. +, positive; -, negative; nd, no data. Data on colony color; cellular size; motility; and the growth of temperature, pH, and NaCl for five type strains were from the studies in [8,9,10,11,12]. The results of the API ZYM and 20NE tests for strains 2 and 3 were from the studies in [11,12].

Characteristics	1	2	3	4	5	6
Colony color	Light-yellow	Yellowish	Orange-red	Beige to pinkish	Orange yellow	nd
Cellular width (μm)	0.4–0.6	0.5–0.6	0.6–1.0	0.4–0.7	0.3–0.7	0.4–0.6
Cellular length (μm)	2.0–2.8	1.7–2.4	1.7–3.2	2.0–3.0	0.9–5.0	1.4–2.3
Motility	+	+	+	+	-	+
Growth of						
Temperature range (°C)	10–40	25–37	20–40	4–45	10–40	10–40
Optimal temperature (°C)	30	30	30	37	30–37	30
pH range	4.0–9.0	6.5–9.5	6.0–9.0	6.0–9.0	5.0–8.0	4.0–10.0
Optimal pH	7.0	7.0–7.5	7.0	7.0–8.0	7.0–8.0	7.0–8.0
NaCl range (*w*/*v*, %)	0–5.0	0.5–5.5	0–6.0	0–3.0	0–6.0	0–6.0
Optimal NaCl (*w*/*v*, %)	1.0	1.5–2.0	2.0	1.0	2.0	0–1.0
Enzymatic activity						
*α*-Glucosidase	-	nd	-	-	+	-
Casein hydrolysis	-	+	nd	+	+	+
Reduction of nitrate to nitrite	+	+	-	+	+	+
Denitrification	-	-	-	+	+	-
D-glucose fermentation	-	-	-	+	+	+
Utilization of						
D-Glucose	+	-	-	+	+	+
D-Mannose	+	nd	-	+	+	+
Potassium gluconate	-	nd	nd	-	+	+

**Table 2 microorganisms-11-02870-t002:** The profiles of cellular fatty acids of strain 3B26^T^ and related reference type strains. Strains: 1, 3B26^T^; 2, *S. zhangzhouensis* FJAT-52072^T^; 3, *S. amazonensis* SB2B^T^; 4, *S. litorisediminis* SMK1-12^T^; 5, *S. jiangmenensis* JM162201^T^. Values in Table 2 represent percentages of total fatty acids. Values below 0.5% and/or 0% in all strains are not shown. tr, trace amount, below 0.5%; nd, not detected. Fatty acid values greater than 10% are highlighted in bold. Values for strains 1 and 3–5 were from this study, while those for strain 2 were from the study in [12].

Fatty Acids	1	2	3	4	5
C_12:0_	4.7	4.0	3.6	4.8	5.9
C_13:0_	2.7	3.6	2.5	2.0	3.1
C_14:0_	2.0	1.8	1.5	1.7	2.4
C_15:0_	4.1	nd	4.7	3.5	5.0
**C_16:0_**	**11.0**	7.3	**10.2**	**12.0**	**11.5**
C_17:0_	3.4	2.6	4.5	4.2	3.6
C_18:0_	0.8	nd	1.3	1.8	0.6
C_11:0_ 3OH	0.9	1.8	1.3	1.4	1.5
C_12:0_ 3OH	2.1	2.3	1.9	3.1	2.8
C_16:0_ 3OH	0.6	nd	0.6	0.7	tr
C_15:1_ *ω*6*c*	tr	1.2	0.9	tr	0.7
C_17:1_ *ω*6*c*	0.9	1.9	1.4	1.0	1.5
**C_17:1_ *ω*8*c***	8.5	**15.1**	**11.1**	7.5	8.5
C_18:1_ *ω*9*c*	2.0	1.5	1.4	1.5	1.5
iso-C_13:0_	3.7	4.6	3.5	3.4	4.2
iso-C_13:0_ 3OH	5.3	4.7	4.0	3.0	3.2
iso-C_14:0_	0.6	1.0	0.5	tr	0.7
**iso-C_15:0_**	**16.5**	**18.9**	**15.7**	**16.6**	8.9
iso-C_15:0_ 3OH	tr	tr	tr	tr	0.5
iso-C_16:0_	1.3	0.6	0.9	1.0	0.6
iso-C_17:0_	2.2	0.8	1.8	2.4	1.1
Summed feature 1 ^#^	0.9	1.0	0.7	0.8	1.7
Summed feature 2 ^#^	0.8	0.7	tr	0.6	2.4
**Summed feature 3 ^#^**	**14.1**	**17.3**	**15.4**	**14.5**	**18.2**
Summed feature 8 ^#^	4.2	4.6	3.5	5.1	4.5

^#^, summed features are fatty acids that cannot be resolved reliably from another fatty acid using the chromatographic conditions chosen. The MIDI system groups are fatty acids grouped together into one feature with a single percentage of the total. Summed feature 1, iso-C_15:1_ H and/or C_13:0_ 3OH; summed feature 2, C_14:0_ 3OH and/or iso-C_16:1_ I; summed feature 3, C_16:1_ *ω*6*c* and/or C_16:1_ *ω*7*c*; summed feature 8, C_18:1_ *ω*6*c* and/or C_18:1_ *ω*7*c*.

**Table 3 microorganisms-11-02870-t003:** The distribution of the key genes and pathways of nitrogen, sulfur, and phosphorus metabolism and melanin biosynthesis in strain 3B26^T^ and its related reference type strains. Strains: 1, 3B26^T^; 2, *S*. *khirikhana* TH2012^T^; 3, *S*. *zhangzhouensis* FJAT-52072^T^; 4, *S*. *amazonensis* SB2B^T^; 5, *S*. *litorisediminis* SMK1-12^T^; 6, *S*. *jiangmenensis* JM162201^T^. +, present; −, absent.

Pathways	K number	Gene	1	2	3	4	5	6
**Nitrogen metabolism**								
Denitrification	K02567	*napA*	+	+	+	+	+	+
	K02568	*napB*	+	+	+	+	+	+
	K00368	*nirK*	+	+	+	+	+	−
	K04561	*norB*	+	+	+	+	+	+
	K02305	*norC*	+	+	+	+	+	+
Dissimilatory nitrate reduction	K02567	*napA*	+	+	+	+	+	+
	K02568	*napB*	+	+	+	+	+	+
	K00362	*nirB*	+	−	−	−	−	−
	K00363	*nirD*	+	−	−	−	−	−
	K03385	*nrfA*	+	+	+	+	+	+
	K15876	*nrfH*	+	+	+	+	+	+
Assimilatory nitrate reduction	K00372	*nasA*	+	−	−	−	−	−
	K00360	*nasB*	+	−	−	−	−	−
**Sulfur metabolism**								
Assimilatory sulfate reduction	K00956	*cysN*	+	+	+	+	+	+
	K00957	*cysD*	+	+	+	+	+	+
	K00955	*cysC*	+	+	+	+	+	+
	K00390	*cysH*	+	+	+	+	+	+
	K00380	*cysJ*	+	+	+	+	+	+
	K00381	*cysI*	+	+	+	+	+	+
**Phosphorus metabolism**								
Two-component system	K02039	*phoU*	+	+	+	+	+	+
	K07636	*phoR*	+	+	+	+	+	+
	K07657	*phoB*	+	+	+	+	+	+
	K08478	*pgtC*	+	+	+	+	+	+
	K08475	*pgtB*	+	+	+	+	+	+
	K08476	*pgtA*	+	+	+	+	+	+
Transporters	K02040	*pstS*	+	+	+	+	+	+
	K02037	*pstC*	+	+	+	+	+	+
	K02038	*pstA*	+	+	+	+	+	+
	K02036	*pstB*	+	+	+	+	+	+
	K03306	*pit*	+	+	+	+	+	+
	K02044	*phnD*	+	+	+	+	+	+
	K02042	*phnE*	+	−	−	−	−	−
	K02041	*phnC*	+	+	+	+	+	+
Organic phosphoester hydrolysis	K01077	*phoA*	+	+	+	+	+	+
	K01113	*phoD*	+	+	+	+	+	+
	K07093	*phoX*	+	+	+	+	+	+
	K09474	*phoN*	−	−	−	−	−	+
**Melanin biosynthesis**	K00500	*phhA*	+	+	+	+	+	+
	K00832	*tyrB*	+	+	+	+	+	+
	K00457	*hppD*	+	+	+	+	+	+

## Data Availability

All relevant data are within the paper and Appendix A.

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
