# Peer review of "Polyphasic Characterization and Genomic Insights into an Aerobic Denitrifying Bacterium, Shewanella zhuhaiensis sp. nov., Isolated from a Tidal Flat Sediment"

_microorganisms, 2023, doi:10.3390/microorganisms11122870_

Round 1

Reviewer 1 Report

Comments and Suggestions for Authors

The manuscript identified a new species of Shewanella by polyphasic characterization and genomic analysis. The author should consider the following issues before accepting the manuscript.

1. Lines 77 and 82. What do these enzymatic activities include? Please specify them and give the results in the results section.

2. Line 132, check the grammar.

3. Line 136 and Figure 1. Please provide the SEM image containing multiple cells or with other shape.

4 Line 144-145, Please provide the image of gram staining.

5. Why the rRNA and tRNA gene numbers are lower than other strains?

Comments on the Quality of English Language

None.

Reviewer 2 Report

Comments and Suggestions for Authors

The work carried out a comprehensive characterization, including both morphological, physiological, and biochemical, as well as genetic analysis of strain 3B26T isolated from tidal flat sediment. The work was carried out at a good methodological level; for the comparative characterization of this strain belonging to the Shewanella genus, 4 other Shewanella species, close in terms of phylogenetic analysis, were used. As a result, they showed unique physiological and biochemical characteristics of strain 3B26T, which do not duplicate any of the compared Shewanella species.

After deciphering the genome of 3B26T, phylogenetic and taxonogenomic Analyses were carried out. It turned out that, in accordance with established standards, strain 3B26T can be isolated as a new species within the genus Shewanella. The authors suggest the name gelatinilytica sp. nov. for this novel member of the genus Shewanella. In this regard, in some section it is worth clearly articulating why gelatin hydrolysis is identified as a unique property for this species from the Shewanella genus.

The authors also studied gene distribution of COG categories for strain 3B26T. The distribution of key genes and pathway of nitrogen, sulfur, and phosphorus metabolism were also studied. In this regard, a good addition to the analysis of deciphered genes would be an attempt to identify candidates responsible for the characteristic light-yellow color of this bacterium. It is possible that there are literature data on pigment color factors for other Shewanella species. In any case, the most phylogenetically close representative, S. khirikhana TH2012T, also has a light-yellow color.

Other comments:

Reword the abstract to 200 words, which is the maximum possible according to the author guidelines. In the current version, your abstract consists of 266 words. It may be possible to partially reduce the details in the description of the strain in the abstract, but it is important to add in which geographic region the 3B26T was isolated.

For a more visual understanding, a geographical map of China should be attached, on which the region where the 3B26T strain was isolated would be marked.
